

# Trajectory classification to support effective and efficient field-road classification

Ying Chen, Kaiming Kuang and Caicong Wu

College of Information and Electrical Engineering, China Agricultural University, Beijing, China
Key Laboratory of Agricultural Machinery Monitoring and Big Data Applications, Ministry of Agriculture and Rural Affairs, Beijing, China

## ABSTRACT

Field-road classification, which automatically identifies in-field activities and out-of-field activities in global navigation satellite system (GNSS) recordings, is an important step for the performance evaluation of agricultural machinery. Although several field-road classification methods based only on GNSS recordings have been proposed, there is a trade-off between time consumption and accuracy performance for such methods. To obtain an optimal balance, it is important to choose a suitable field-road classification method for each trajectory based on its GNSS trajectory quality. In this article, a trajectory classification task was proposed, which classifies the quality of GNSS trajectories into three categories (high-quality, medium-quality, or low-quality). Then, a trajectory classification (TC) model was developed to automatically assign a quality category to each input trajectory, utilizing global and local features specific to agricultural machinery. Finally, a novel field-road classification method is proposed, wherein the selection of field-road classification methods depends on the trajectory quality category predicted by the TC model. The comprehensive experiments show that the proposed trajectory classification method achieved 86.84% accuracy, which consistently outperformed current trajectory classification methods by about 2.6%, and the proposed field-road classification method has obtained a balance between efficiency and effectiveness, *i.e.*, sufficient efficiency with a tolerable accuracy loss. This is the first attempt to examine the balance problem between efficiency and effectiveness in existing field-road classification methods and to propose a trajectory classification specific to these methods.

# INTRODUCTION

Although several field-road classification methods based solely on global navigation satellite system (GNSS) recordings of agricultural machinery have been proposed, they often suffer from either a performance robustness issue or a time consumption problem. The term "performance robustness" refers to the stability of classification performance across various input trajectories, while "time computation" denotes the time required to run a classification method (*Chen et al., 2021*; *Poteko, Eder & Noack, 2021*; *Chen et al., 2022*;

Corresponding authors
Ying Chen, chenying@cau.edu.cn
Caicong Wu, wucc@cau.edu.cn

*Zhang et al., 2022*). Given that practical applications require both effective and efficient field-road classification, it is necessary to address these two issues.

There are two typical field-road classification methods: DBSCAN + Rules and graph convolutional neural network (GCN). DBSCAN + Rules uses a density-based clustering method, *i.e.*, density-based spatial clustering of applications with noise (DBSCAN), to detect the categories of all points in a trajectory, and then utilizes rules to correct the clustering result based on the parallel direction characteristic (*Chen et al., 2021*). Since DBSCAN + Rules relies much on DBSCAN, it does not perform robustly because of the unstable effectiveness performance of DBSCAN, and yet it has a pretty low computation time cost due to the efficiency of DBSCAN. That is, if trajectories are high-quality (*e.g.*, uniform density, regular direction, and clear boundaries), promising performance can be expected. In contrast, there is a significant drop in performance on lower-quality trajectories. To address the performance robustness problem, a deep learning method known as GCN was introduced (*Chen et al., 2022*). The method constructs a spatio-temporal graph based on spatial and temporal relationships between points in an input trajectory and then employs the graph convolution process to extract spatio-temporal features for supporting field-road classification. Although the GCN method exhibits robust performance due to its powerful spatio-temporal features, it incurs a significant computation time cost (*e.g.*, 48 h for 28,025 machines). The time cost mainly comprises the time needed to construct the spatio-temporal graph and train the complex neural network. Meanwhile, two additional field-road classification methods also have been proposed. One method involves utilizing decision trees (DT) combined with features that reflect motion characteristics (*Poteko, Eder & Noack, 2021*). However, in comparison to DBSCAN + Rules and GCN, the DT method's performance is limited, as neither the features nor the DT is powerful. The other is DBSCAN+OD+DBI (DBSCAN plus object detection and Davis Bouldin index), which was proposed to enhance the robustness of DBSCAN (*Zhang et al., 2022*), but it still does not reach the effectiveness achieved by GCN. As a result, DBSCAN + Rules and GCN are selected in this article. To leverage both the efficiency of DBSCAN + Rules and the effectiveness of GCN simultaneously, a selection strategy should be implemented. This strategy would select the most suitable field-road classification method for an input trajectory based on its quality. However, the prerequisite for this strategy is the availability of a trajectory classification method that is capable of automatically classifying trajectories according to their qualities. In practice, both the definition and the method of trajectory quality classification are under-investigated.

Although several trajectory classification tasks have been presented in recent years, most of them focus on transportation mode classification (*Zheng et al., 2009*; *Yuan et al., 2010*; *Zheng, 2015*; *Bian et al., 2019*; *Wang et al., 2020*; *Zeng et al., 2023*) or ship movement classification (*Sheng et al., 2018*; *Chen et al., 2020*). For example, the work of *Xiao et al. (2017)* was to classify the transportation mode in a trajectory, such as walking, bike, bus, driving, or train, and the study of *Jiang et al. (2023)* was to classify the high-speed travel modes (*i.e.*, bus, car, and railway) in the sequence. It is important to note that transportation mode classification and trajectory quality classification address distinct facets of trajectory analysis. Consequently, a classification task specific to trajectory quality is required.

Furthermore, many automatic trajectory classification methods (*Dodge, Weibel & Forootan, 2009*; *Wang et al., 2020*; *Klich, Ecochard & Subtil, 2021*) that utilize machine learning methods to assign a category to an input trajectory have been developed for transportation mode classification tasks. However, they do not fit our trajectory quality classification task. These trajectory classification methods attempted to capture movement characteristics either from a global scope which represents the entire trajectory or from a local scope which focuses on specific segments. As a segment constitutes only a part of a trajectory, trajectory segmentation, which divides a trajectory into segments, should be conducted before extracting features based on those segments. For example, in *Dodge, Weibel & Forootan (2009)*, local features were extracted for each segment after slicing a trajectory into segments using a fixed time interval, and in *Xiao et al. (2017)*, both global and local features were extracted to represent a trajectory based on several movement-based parameters (*e.g.*, speed, acceleration, turn angle, and sinuosity). These movement-based features have been proven to be helpful for classifying transportation modes. However, these features are not sufficient for the quality classification of agricultural machinery trajectories, as both movement-based and point-density-based characteristics play important roles (*Chen et al., 2021*; *Chen et al., 2022*; *Zhang et al., 2022*). So, when dealing with the trajectory quality classification task, a dedicated feature extraction method is essential.

To support efficient and effective field-road classification, this article aims to develop a well-performed trajectory classification method. Firstly, we established a trajectory classification task for two state-of-the-art field-road classification methods: DBSCAN + Rules and GCN. This task involved categorizing trajectories into three distinct qualities (high-quality, medium-quality, and low-quality), and its quality evaluation criterion was a combination of point density, driving direction, and the clarity of field boundaries within each trajectory. Subsequently, an automatic trajectory classification (TC) model was developed. This model extracted global and local features based on movement-based and point-density-based parameters. Finally, a hybrid field-road classification method, namely TCModel-based DBSCAN+GCN field-road classification, was proposed. In this classification method, DBSCAN + Rules was utilized exclusively for high-quality trajectories, while the GCN method was applied specifically to medium-quality trajectories.

Our main contributions can be summarized as follows: (1) To our best knowledge, it is the first attempt to investigate the balance problem of the efficiency and effectiveness of existing field-road classification methods. To achieve a good balance, we designed the trajectory classification task and developed an automatic trajectory classification model to support the selection of appropriate field-road classification methods for input trajectories. Moreover, to guarantee the efficiency and effectiveness of the whole process including trajectory classification and field-road classification, our trajectory classification model itself also has a good balance of efficiency and effectiveness. (2) To achieve accurate trajectory classification, we extracted global and local features reflecting the movement and point-density characteristics of agricultural machinery trajectories.

The article is organized as follows. Firstly, the data used in our research is described, the trajectory classification task is defined, and the trajectory classification method is

**Table 1  The abbreviations and their corresponding terminologies.**

| Abbreviations | Terminologies |
|---|---|
| DBSCAN | Density-based spatial clustering of applications with noise |
| DBSCAN + Rules | DBSCAN plus direction distribution-based rules |
| DBSCAN + OD + DBI | DBSCAN plus Object Detection and Davis Bouldin index |
| DT | Decision Tree |
| FRCEFfct data | Field-road classification effectiveness evaluation data |
| FRCEFfcy data | Field-road classification efficiency evaluation data |
| GCN | Graph Convolutional Neural network |
| GNSS | Global Navigation Satellite System |
| RF | Random Forest |
| TC | Trajectory Classification |
| TCRule-based DBSCAN + GCN | A hybrid field-road classification using the DBSCAN + Rules and GCN method according to the result of rule-based trajectory classification |
| TCModel-based DBSCAN + GCN | A hybrid field-road classification using the DBSCAN + Rules and GCN method according to the result of the trajectory classification model |

presented. Then, experimental results along with an in-depth analysis are provided. Finally, the conclusion of our contributions and future research are presented.

# MATERIALS AND METHODS

In this section, we will begin by summarizing the dataset used in our study. We will first introduce the task definition of our trajectory classification, and present our trajectory classification model. Then, we will provide a detailed explanation of our hybrid field-road classification method. Finally, we will present the experiments of our automatic trajectory classification and field-road classification methods. Moreover, alphabetically sorted related abbreviations and their corresponding terminologies are provided in Table 1.

## Dataset

The data used in this article were selected from the Precision Agriculture Application Project Data Service Platform, which receives real-time GNSS recordings of agricultural machinery from various agricultural machinery manufacturers (*Wu et al., 2023*). The acquisition accuracy (*i.e.,* position accuracy) of the used GNSS receivers is about 5 m (CEP, Circular Error Probable). For each point in the trajectories, there are five recorded parameters: timestamp, longitude, latitude, speed, and direction. According to different evaluation purposes, three GNSS trajectory datasets are constructed as follows.

Trajectory classification evaluation data (the TC data): it includes 1,005 daily trajectories collected from 939 harvesters, in which 401 trajectories were recorded during wheat harvesting from May 2021 to June 2021 and 604 trajectories were recorded during paddy harvesting from July 2021 to November 2021. The dataset is used to evaluate trajectory classification.

Field-road classification effectiveness evaluation data (the FRCEFfct data): it consists of the wheat harvester trajectory dataset and paddy harvester trajectory dataset used in the research of *Chen et al. (2022)*. The wheat harvester trajectory dataset contains 150 daily wheat harvester trajectories collected by Weichai Lovol Heavy Industry Co. from Jun. 2021 to Jul. 2021. The paddy harvester trajectory dataset includes 100 paddy harvester trajectories collected by Jiangsu World Agriculture Machinery Co. from Oct. 2021 to Nov. 2021. Moreover, the logging interval setting of GNSS receivers is 5 s and 30 s for the wheat and paddy harvester trajectories, respectively. The dataset is used to evaluate the effectiveness of field-road classification methods.

Field-road classification efficiency evaluation data (the FRCEFfcy data): it is a large-scale dataset containing 28,025 daily trajectories which were collected by 28,025 wheat harvesters during harvesting season, *i.e.*, from 12:00 pm Jun.15 to 11:59 am Jun.16 in 2022. Moreover, regarding the logging interval, 54.05% of GNSS receivers were set to collect data every 5 s, and 45.95% of GNSS receivers were set to collect data every 30 s. The data is used to evaluate the efficiency of field-road classification methods in a practical scenario that involves performing large-scale trajectory analysis for agricultural machines distributed across multiple regions in China.

## Task definition of trajectory classification

In practice, due to signal loss and signal interference, GNSS receivers used to collect the trajectories of agricultural machinery do not always work exactly as their settings, which leads to trajectories with various qualities. Furthermore, trajectory quality often has a great impact on the effectiveness of existing field-road classification methods. For example, DBSCAN + Rules is an efficient field-road classification method. However, it works well only on high-quality trajectories (*Chen et al., 2021*). On the contrary, GCN is an effective field-road classification method, *i.e.*, its classification accuracy would not drop much even on medium-quality trajectories (*Chen et al., 2022*). However, the method often consumes a lot of computation time. Moreover, in the worst case, some trajectories should be discarded because they are too bad to distinguish field and road points even for human annotators. Therefore, to help develop a field-road classification with a good balance of efficiency and effectiveness, we classify trajectories into the following three categories according to their qualities.

- High-quality trajectory (*e.g.*, Fig. 1A): it should satisfy the following three shape characteristics:
  1. uniform density: the time interval between two successive points is almost fixed, which indicates that point density remains uniformly in field;
  2. regular direction: directions while driving in field and on road are usually regular, which means the directions of two successive points seldom change rapidly except in the case of normal operations, such as turning, avoiding obstacles, and so on;
  3. clear boundary: there is a clear boundary for a field, which facilitates the easy distinction between field and road points. This clarity is helpful for both field-road classification methods and human annotators.

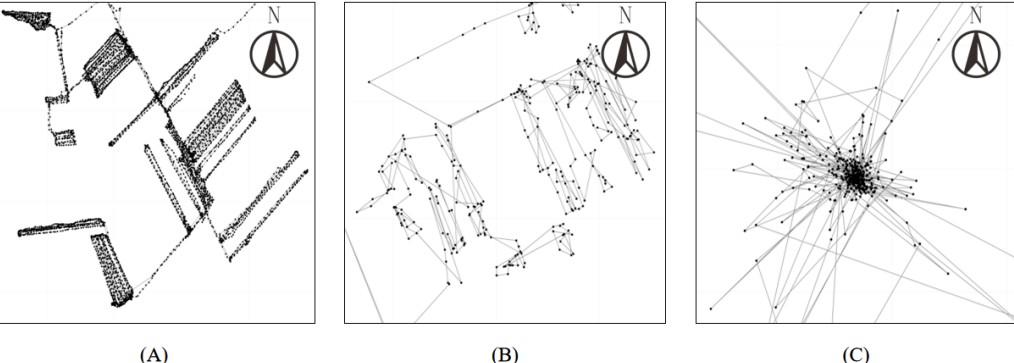

**Figure 1  Trajectory examples with different quality categories.** (A) A high-quality trajectory with uniform density, regular direction, and clear boundary; (B) a medium-quality trajectory with relative uniform density, relative regular directions, and curved/intermittent boundary; (C) a low-quality trajectory with variable density, irregular direction, and unclear boundary.

In practice, the DBSCAN + Rules method usually works well for such high-quality trajectories.

- Medium-quality trajectory (*e.g.*, Fig. 1B): it should satisfy the following shape characteristics:
  1. relative uniform density: unlike high-quality trajectories, the point density of the medium-quality trajectory is not uniformly distributed. In other words, the time interval between two successive points is not always fixed;
  2. relative regular direction: most directions in the trajectory are regular. In other words, the directions of two successive points sometimes change suddenly, and such changes are primarily caused by GNSS signal drift;
  3. curved or intermittent boundary: the borders of fields are non-linear, curved, or sporadically broken. Despite these complexities, human annotators can identify such fields with additional effort.

In practice, the GCN method often works well on such medium-quality trajectories because the method has lower quality requirements for trajectories.

- Low-quality trajectory (*e.g.*, Fig. 1C): it has the following shape characteristics:
  1. variable density: points in the trajectory are often discontinuous, which leads to a variable point density in the trajectory;
  2. irregular direction: directions in the trajectory are disordered;
  3. unclear boundary: the edges of fields are indistinct or entirely missing, complicating the distinction between field and non-field areas.

It is nearly impossible to identify fields and roads in such a low-quality trajectory even by human annotators, not to mention automatic field-road classification methods, *e.g.*, DBSCAN + rules and the GCN method.

Trajectory quality is an overall evaluation based on point density, driving direction, and the clarity of each field boundary in the trajectory. As a result, the trajectory quality evaluation is subjective. To avoid the subjective influence, two human annotators

independently annotated each trajectory in the TC data. If their annotations were consistent, the annotations were approved. In cases where the annotations were not consistent, a third human annotator was involved to provide a final annotation. Such human-annotated data serves as ground truth data for the trajectory classification task.

Moreover, the three shape characteristics are determined by the practical logging interval and acquisition accuracy of GNSS receivers. Specifically, "density" is mainly determined by the practical logging intervals. For example, in the wheat harvester trajectory dataset of the FRCEFfct data, although the logging interval setting of GNSS receivers was 5 s, 16.51% of points were recorded not by 5 s. "Direction" largely relies on the acquisition accuracy. However, since the GNSS receivers used in our data have low accuracy, 5 m (CEP), the recording is often not very straight even when the focused agricultural machine drives straight. Lastly, "boundary" depends on both the practical logging interval and acquisition accuracy.

## Automatic trajectory classification
### Overview

As shown in Fig. 2, the proposed automatic trajectory classification method consists of three parts: data preprocessing, feature extraction, and random forest (RF)-based classification.

- Data preprocessing: it provides more parameters for each trajectory to support its feature extraction.
- Feature extraction: it includes global feature extraction and local feature extraction.

  - Global feature extraction: it extracts 123 features based on the overall characteristics of a trajectory, such as direction change, time difference, and so on.
  - Local feature extraction: it extracts 52 features based on the characteristics of a segment, where a segment is part of a trajectory.

- RF-based classification: based on $175(=123+52)$ features, a tree-based classification method, random forest (RF), is applied to develop a trajectory classification model. Such a model can identify the category of a trajectory ("high-quality", "medium-quality" or "low-quality").

### Data preprocessing

Based on the five recorded parameters of a GNSS point, data preprocessing was carried out to obtain more useful parameters. Specifically, each GNSS point in our dataset is represented by a fourteen-parameter vector including longitude, latitude, direction, speed, acceleration, jerk, bear rate, longitude difference, latitude difference, direction difference, time delta, sinuosity, distance, and vin. The former four parameters (longitude, latitude, direction, and speed) are directly recorded, and the latter nine parameters are derived. Moreover, jerk, bear rate, longitude difference, and latitude difference illustrate the relationship between the current point and its predecessor; time delta represents the time interval between two consecutive points; sinuosity measures the curvature between two successive paths; distance denotes the actual distance between two successive points, and vin is calculated by dividing distance by time delta (*Chen et al., 2022*).

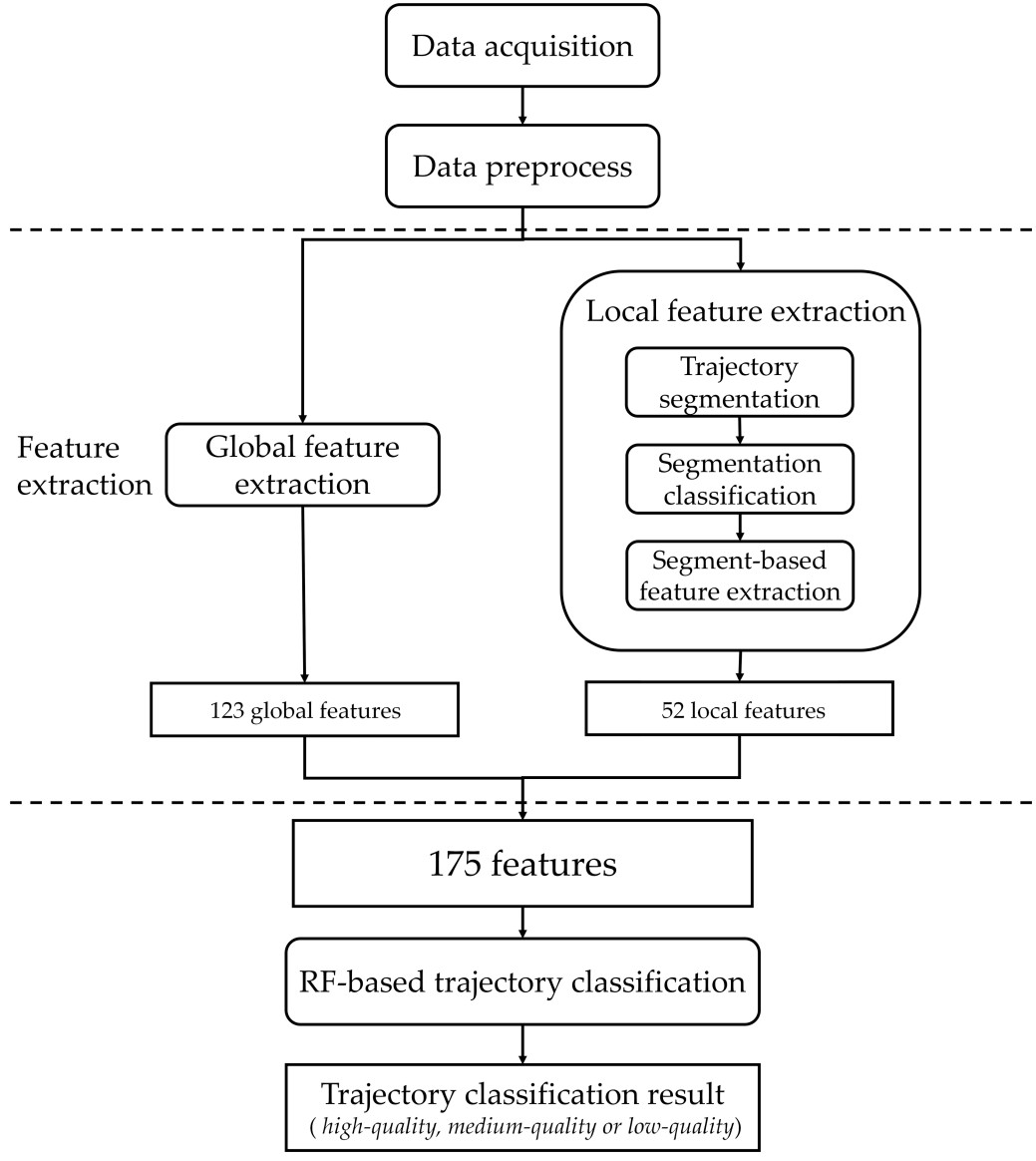

**Figure 2**   **The workflow of the proposed automatic trajectory classification method.**

### Global feature extraction

Similar to the previous work (*Xiao et al., 2017*), 123 global features are extracted to represent a trajectory, where global features are the statistics of all points in a trajectory. As shown in Table 2, 17 statistical measures are used, seven parameters are selected, and then a statistical value is calculated per parameter and statistical measure (*Mardia & Jupp, 1999*). Besides the four parameters (speed, sinuosity, acceleration, and direction difference) used in *Xiao et al. (2017)*, additional three parameters are selected: distance, time delta, and vin. Distance and time delta can reflect point-density-based characteristics, and vin is indicative of movement-based characteristics (*Xiao et al., 2017*). Then, as the study of *Zheng et al. (2008)* pointed out kinetic parameters can enhance the robustness of trajectory

**Table 2  A summary of the global features and local features (an asterisk (\*) indicates parameters used in previous works).**

| Type | Feature description | Number |
|---|---|---|
| Global features | Parameters (7): speed\*, sinuosity\*, acceleration\*, direction_diff\*, distance, time_delta, vin | $7 \times 17 = 119$ |
| | Statistics (17): mean, standard deviation, mode, maximum three values, minimum three values, ranges of value, quartile, interquartile range, skewness, kurtosis, coefficient of variation, autocorrelation coefficient | |
| | Parameters (4): heading change rate\*, stop rate\*, velocity change rate\*, total distance | 4 |
| Local features | Parameters (6): direction delta, distance, speed, sinuosity, bear rate delta, vin delta | $6 \times 2 \times 4 = 48$ |
| | Statistics (2): mean, standard deviation | |
| | Parameters (4): heading change rate\*, stop rate\*, velocity change rate\*, total distance | $1 \times 4 = 4$ |

classification models, in addition to the three kinetic parameters (heading change rate, stop rate, and velocity change rate) used in *Xiao et al. (2017)*, an additional kinetic parameter, total distance, is employed to capture point-density-based characteristics.

### Local feature extraction

Inspired by the work of *Dodge, Weibel & Forootan (2009)*, we designed a local feature extraction method specifically for agricultural machinery, which extracts 52 local features to enhance the feature representation of a trajectory. As illustrated in Fig. 2, the local feature extraction consists of three steps: trajectory segmentation which provides a set of segments, segment classification which automatically classifies segments into four categories, and segment-based feature extraction which extracts features based on the segments and their respective categories.

Firstly, three trajectory segmentation methods are conducted separately, and each of them slices a trajectory into a sequence of segments. As these methods have varying strengths and weaknesses in extracting characteristics, all of them are employed in this study.

- Segmentation based on fixed time interval: a trajectory is sliced into segments with a fixed time interval.
- Segmentation based on fixed point number: a trajectory is split into segments with a fixed number of points.
- Segmentation based on spare time: if the time interval between two successive points is more than a threshold (*e.g.*, 600 s), a split is made in the middle of the two points.

Secondly, according to the two kinds of GNSS signal qualities (strong and weak/noisy) and the two distinct movement behaviors (driving straight and making a turn), the segments are classified into the following four categories to reflect the combinations of two crucial characteristics: point density and driving direction.

- Low direction change and low point density: it represents the case that agricultural machinery drives straight normally yet with weak/noisy GNSS signals.
- Low direction change and high point density: it represents the case that agricultural machinery drives straight normally with strong GNSS signals.
- High direction change and low point density: it represents the case that agricultural machinery makes a turn normally yet with weak/noisy GNSS signals.
- High direction change and high point density: it represents the case that agricultural machinery makes a turn normally with strong GNSS signals.

Finally, as detailed in Table 2, 12 local features are extracted per category. Firstly, six parameters (direction delta, distance, speed, sinuosity, bear rate delta, and vin delta) and two statistical measures (mean and standard deviation) are selected. Then, for each segment, a statistical value is calculated for each parameter and measure. Finally, for each category, the mean of the statistical values of segments belonging to that category are computed.

## TCModel-based field-road classification

In Section 'Introduction', the efficiency and effectiveness of either field-road classification method (DBSCAN + Rules or GCN) were highlighted. To take advantage of both methods, we proposed a hybrid field-road classification method based on trajectory classification (TC) models, namely TCModel-based DBSCAN+GCN (a hybrid field-road classification using the DBSCAN + Rules and GCN method according to the result from the trajectory classification model), in which the application of either DBSCAN + Rules or GCN depends on the quality category predicted by the TC model. Specifically, for each given trajectory, the developed trajectory classification model is employed to determine its quality category. Subsequently, a field-road classification method is applied depending on the quality category: DBSCAN + Rules is used for high-quality trajectories; the GCN method is applied to medium-quality trajectories; for low-quality trajectories, all points are categorized as "road".

## Method implementation

For the trajectory classification method, the random forest classifier directly used the implementation in Scikit-learn (*Pedregosa et al., 2011*). For the four existing field-road classification methods, including DBSCAN + Rules, DT, DBSCAN + OD +DBI, and GCN, we directly adopted the implementations and settings as specified in their original articles (the code is available at https://github.com/Agribigdata/Field_road_dataset) (*Chen et al., 2021*; *Poteko, Eder & Noack, 2021*; *Zhang et al., 2022*; *Chen et al., 2022*). Moreover, other parts of our trajectory classification and TCModel-based field-road classification were implemented in Python language.

## RESULTS AND DISCUSSION

In this section, we conduct two types of evaluations. One is to assess the effectiveness of our trajectory classification method, and the other is to evaluate the performance of our TCModel-based field-road classification method.

**Table 3  The overall performance trajectory classification using different feature extraction methods.**

| Method | P | R | F1 | ACC |
|---|---|---|---|---|
| Original GF | 82.97 | 78.72 | 80.62 | 84.07 |
| Original GF+LF | 83.99 | 78.74 | 81.05 | 84.78 |
| Overall GF | 86.22 | 84.92 | 85.50 | 85.97 |
| Overall GF+LF | **86.77** | **85.85** | **86.24** | **86.67** |

**Notes.**

P, Precision score; R, Recall score; F1, F1-score; Acc, accuracy score.
The numbers in bold indicate the best performances.

## Automatic trajectory classification evaluation
### Experimental settings
**Baseline methods**: to demonstrate the effectiveness of our feature extraction methods, a comparison was made between our proposed feature extraction features and two existing feature extraction methods that have been commonly used in the transportation mode classification studies: original global feature extraction and original global+local feature extraction (*Dodge, Weibel & Forootan, 2009*; *Xiao et al., 2017*; *Meng et al., 2020*; *Hechifa et al., 2023*).

1. Original global feature extraction (Original GF): it follows the work of *Dodge, Weibel & Forootan (2009)* to extract 71 features based on four parameters (speed, sinuosity, acceleration, and direction_diff), as shown in the global feature part of Table 2;

2. Original global+local feature extraction (Original GF+LF): it follows the work of *Xiao et al. (2017)* to extract 119 features including the 71 original global features and 48 local features in Table 2;

3. Overall global feature extraction (Overall GF): it is our global feature extraction including the 123 global features in Table 2.

4. Overall global+local feature extraction (Overall GF+LF, our feature extraction): it extracts 175 global+local features including the 123 global features and 52 local features in Table 2.

**Model training and evaluation**: the experiment of trajectory classification consists of three steps: data split, model training, and model test. The TC data was randomly split into 80% and 20% for training and testing, the training data was used to train a trajectory classification model, and the test data was used to evaluate the performance of the developed model. Similar to the previous work on a classification problem (*Chen et al., 2021*; *Chen et al., 2022*; *Zhang et al., 2022*), four metrics, *i.e.,* precision, recall, F1-score, and accuracy, were used to evaluate model performance.

### Method comparisons
Table 3 lists the performances of trajectory classification using the four feature extraction methods. Compared to the baselines, our proposed feature extraction (*i.e.,* overall global+local features) performs best, achieving 86.24% in F1-score and 86.67% in accuracy. This indicates that the design of our feature extraction is effective for the quality classification of agricultural machinery trajectories.

**Table 4** The performance across quality categories for trajectory classification using different feature extraction methods.

| Method | Quality | | | | | | | | |
|---|---|---|---|---|---|---|---|---|---|
| | High | | | Medium | | | Low | | |
| | P | R | F1 | P | R | F1 | P | R | F1 |
| Original GF | 86.39 | 92.12 | 89.16 | 78.30 | 68.73 | 73.20 | 84.21 | 75.29 | 79.50 |
| Original GF+LF | 86.46 | **93.20** | **89.71** | 80.37 | 68.89 | 74.19 | 85.14 | 74.12 | 79.25 |
| Overall GF | 88.93 | 83.21 | 85.97 | 83.38 | 88.93 | 86.07 | 86.36 | 82.61 | 84.44 |
| Overall GF+LF | **89.70** | 83.94 | 86.72 | **84.04** | **89.55** | **86.71** | **86.57** | **84.06** | **85.29** |

**Notes.**
The numbers in bold indicate the best performances.

Furthermore, performance improvement mainly comes from incorporating global features specific to agricultural machinery. As illustrated in Table 3, compared to "original global features", the adding of the specific global features in "overall global features" yields 4.88% and 1.9% increase in F1-score and accuracy, respectively. This confirms the effectiveness of additional global parameters in our feature extraction. Moreover, the incorporation of local features also can enhance trajectory classification. For example, compared to "overall global features", "overall global+local features" yield 0.74% and 0.7% improvement in F1-score and accuracy, respectively. This shows that certain characteristics of agricultural machinery may be discernible only within particular segments of a trajectory. In summary, the effective extraction of global and local features specific to agricultural machinery leads to a promising trajectory classification model.

Finally, Table 4 lists the performances of these trajectory classification methods for each quality category. We can observe that our feature extraction method improves the performance of the "medium-quality" and "low-quality" classes, but hurts the performance of the "high-quality" class. For example, comparing "overall global features" to "original global features", the F1-score increases by 12.87% on the "medium quality" class and 4.94% on the "low-quality" class, but decreases by 3.19% on the "high-quality" class. The results demonstrate that the additional features used in our global feature extraction method significantly enhance the representation of medium-quality data, and moderately improve low-quality data, but bring a negative impact on the representation of high-quality data. Comparing "overall global+local features" to "overall global features", the F1-score increases by 0.75% on the "high-quality" class, 0.64% on the "medium quality" class, and 0.80% on the "low-quality" class. This demonstrates that the additional local feature in our method can contribute to a consistent improvement in data representation. In summary, these additional features enhance the overall performance, which validates the effectiveness of our feature extraction method.

## Efficiency and effectiveness evaluation of field-road classification
### Experimental settings
**Baseline methods**: to show the efficiency and effectiveness of the proposed TCModel-based DBSCAN+GCN, it was compared with four state-of-the-art field-road classification methods, DBSCAN + Rules (*Chen et al., 2021*), DT (*Poteko, Eder & Noack, 2021*), DBSCAN

+ OD + DBI (*Zhang et al., 2022*), and GCN (*Chen et al., 2022*), and another hybrid field-road classification method, TCRule-based DBSCAN+GCN (a hybrid field-road classification using the DBSCAN + Rules and GCN method according to the result from rule-based trajectory classification). TCRule-based DBSCAN+GCN uses rule-based trajectory classification as follows: if the majority of the logging intervals in a trajectory are not greater than 5 s, DBSCAN + Rules is selected; otherwise, the GCN method is used. Among the four baseline field-road classification methods, in terms of effectiveness, the GCN method demonstrates outstanding performance in both the wheat harvester trajectory dataset and the paddy harvester trajectory dataset, surpassing the other three methods. In terms of efficiency, the DBSCAN method proves to be significantly more time-efficient, compared to the other three methods.

**Model training and evaluation**: the following two experiments were carried out to evaluate the effectiveness and efficiency of six field-road classification methods, respectively.

**Effectiveness evaluation experiment:** except TCRule-based DBSCAN+GCN, other five field-road classification methods (DBSCAN + Rules, DT, DBSCAN + OD + DBI, GCN, TCModel-based DBSCAN+GCN) were conducted independently on the FRCEFfct data. Firstly, the wheat harvester trajectory dataset (or the paddy harvester trajectory dataset) was randomly split into 80% and 20% for training and testing. Then, the field-road classification method was used to train a model with the training data, and the effectiveness of the developed model was evaluated on the test data. Notice that the trajectory classification model used in TCModel-based DBSCAN+GCN already had been trained with all trajectories in the TC data. Consistent with prior research (*Chen et al., 2021*; *Chen et al., 2022*; *Zhang et al., 2022*), four metrics (*i.e.,* precision, recall, F1-score, and accuracy) were used to evaluate model performances. Moreover, the ignoring of TCRule-based DBSCAN+GCN is attributed to the characteristics of the FRCEFfct data. Specifically, for the wheat harvester trajectories, the majority of the logging intervals are less than 5 s, and only the DBSCAN + Rules method is used. Meanwhile, for the paddy harvester trajectories, the majority of the logging intervals are greater than or equal to 5 s, only GCN is used.

**Efficiency evaluation experiment:** all six field-road classification methods were conducted independently using both the FRCEFfct data and the FRCEFfcy data. In general, for each field-road classification method, there are two steps: training and testing. In the training phase, a field-road classification model was developed using the wheat harvester dataset within the FRCEFfct data, and in the testing phase, the model's time consumption was evaluated on the FRCEFfcy data, where ''time consumption'' is defined as the total time interval from the initiation to the completion of the testing phase, including processes such as data loading, execution of field-road classification, and storage of classification results. Moreover, for the TCModel-based DBSCAN+GCN, the time consumption associated with trajectory classification was taken into account, which refers to the total time required for predicting trajectory quality.

**Table 5** The overall performance of different field-road classification methods on the wheat harvester trajectory dataset.

| Method | P | R | F1 | ACC |
|---|---|---|---|---|
| DBSCAN + Rules | 77.84 | 65.50 | 68.31 | 84.65 |
| DT | 75.16 | 64.31 | 66.79 | 84.23 |
| DBSCAN + OD + DBI | 87.49 | 60.51 | 61.82 | 81.62 |
| GCN | **80.66** | **72.92** | **75.14** | **86.96** |
| TCModel-based DBSCAN+GCN | 80.10 | 72.62 | 74.75 | 86.79 |

**Notes.**
The numbers in bold indicate the best performances.

**Table 6** The overall performance of different field-road classification methods on the paddy harvester trajectory dataset.

| Method | P | R | F1 | ACC |
|---|---|---|---|---|
| DBSCAN + Rules | 67.03 | 67.97 | 66.79 | 71.60 |
| DT | 71.55 | 67.33 | 68.29 | 76.41 |
| DBSCAN + OD + DBI | 70.05 | 64.01 | 64.11 | 72.24 |
| GCN | 82.40 | 79.13 | 80.32 | 84.55 |
| TCModel-based DBSCAN+GCN | **83.04** | **79.21** | **80.50** | **84.82** |

**Notes.**
The numbers in bold indicate the best performances.

## Method comparisons

**Effectiveness evaluation**: Tables 5 and 6 list the performances of the five field-road classification methods on the wheat and paddy harvester trajectory datasets in the FRCEFfct data, respectively. When the dataset is altered, such as switching from the wheat harvester trajectory dataset to the paddy harvester trajectory dataset, there is a notable change in the performance of the same field-road classification method. Among the five methods, the performance of GCN decreases slightly, and the performance of DBSCAN + Rules drops a lot, which confirms the robustness of the GCN method. The observed variations in the performance of the same field-road classification method can be primarily attributed to the alterations in logging interval settings of GNSS receivers used during data collection (*Chen et al., 2022*). For example, TCModel-based DBSCAN+GCN achieves 86.79% accuracy on the wheat harvester trajectory dataset whose logging interval setting is 5 s, and obtains 84.82% accuracy on the paddy harvester trajectory dataset whose logging interval setting is 30 s. The prolonged logging interval typically results in low-quality trajectories, which further affect the performance of field-road classification. So, the quality of the trajectories is a critical factor in determining the effectiveness of field-road classification methods.

Moreover, the proposed TCModel-based DBSCAN+GCN achieves acceptable performance. Specifically, on the wheat harvester trajectory dataset, TCModel-based DBSCAN+GCN surpasses DBSCAN + Rules, DT, DBSCAN + OD +DBI, and by 2.14%, 2.56% and 5.17% in accuracy, respectively. However, compared to GCN, it has a 0.17% drop in accuracy. Similarly, on the paddy harvester trajectory dataset, compared to DBSCAN + Rules, DT, DBSCAN + OD + DBI, and GCN, TCModel-based DBSCAN+GCN achieves

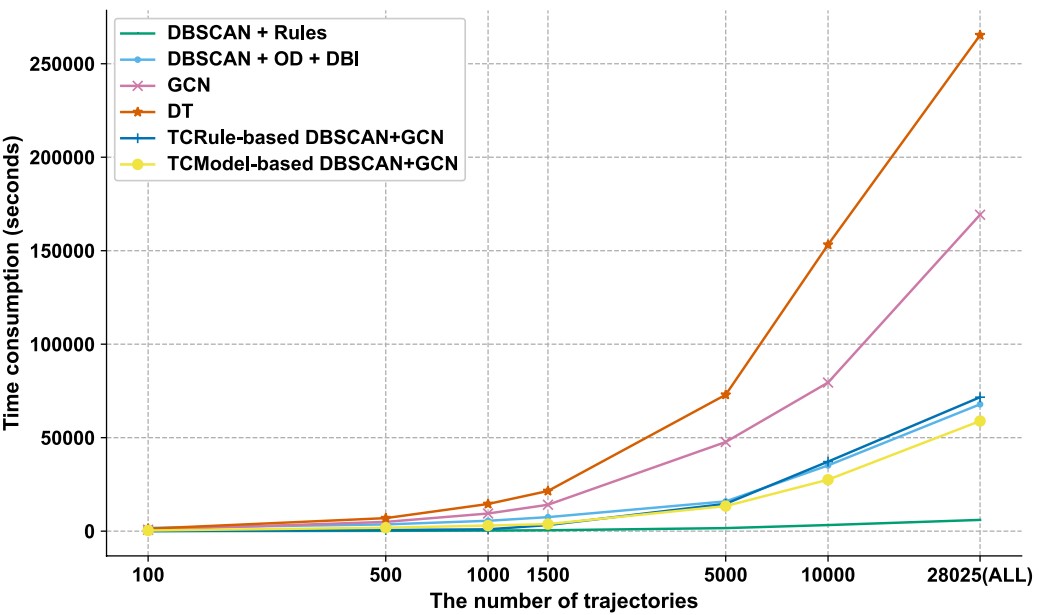

**Figure 3** The time consumption of different field-road classification methods on the FRCEFfcy dataset.

13.22%, 8.41%, 12.58%, and 0.27% increase in accuracy, respectively. These results confirm the effectiveness of our TCModel-based DBSCAN+GCN, which is a hybrid approach that integrates the DBSCAN + Rules and GCN methods.

**Efficiency evaluation**: Figure 3 shows the time consumption of the six field-road classification methods inputted with a different number of trajectories, where the input trajectories were randomly selected from the FRCEFfcy dataset.

As the number of input trajectories increases, the time consumption increases continuously for all methods, where DBSCAN+Rules increases most slowly, and DT increases fastest. Among the whole process of a given field-road classification method, feature extraction is the most time-consuming step. However, the time does not increase linearly with the number of input trajectories due to varying points per trajectory, resulting in a nonlinear time cost in feature extraction.

Additionally, the TCModel-based DBSCAN+GCN displays a distinct advantage when processing over 5,000 trajectories. In a scenario where all trajectories in the FRCEFfcy dataset are utilized, the time consumption of TCModel-based DBSCAN+GCN amounts to approximately 83% of that required by TCRule-based DBSCAN+GCN. This finding substantiates the superiority of our trajectory classification model over the methods relying solely on logging interval settings in terms of field-road classification method selection.

## DISCUSSION

This is the first attempt to define a trajectory classification task specific to the field-road classification. Our automatic trajectory classification approach, which employs a novel

global and local feature extraction method, outperforms the one using the original features by a notable increase (5.19%) in the F1-score and a significant improvement (1.89%) in accuracy. The additional 103 features used in our approach can highlight more characteristics of trajectory quality. For example, the parameters used in our proposed global features (vin and distance) directly indicate point density and indirectly reflect driving direction. In high-quality trajectories, changes in vin (or in distance) tend to be consistent, and conversely, in low-quality trajectories, these parameters exhibit greater variability. Moreover, the parameters of our proposed local features (the heading change rate and the stop rate) are vital factors during the quality classification of agricultural machinery trajectories. In high-quality trajectories, especially during in-field operations, these parameters tend to be stable. Conversely, in low-quality trajectories, they exhibit significant variations.

To demonstrate the effectiveness of our TCModel-based DBSCAN+GCN method, Fig. 4 presents a comparison of field-road classification methods across various quality trajectories. For high-quality trajectories, as depicted in Figs. 4A–E, both the DBSCAN + Rules and GCN methods attain excellence in line with expectations, making them both suitable for high-quality trajectories. For medium-quality trajectories, the DBSCAN + Rules method tends to misclassify "road" as "field", reducing its effectiveness compared to the GCN method, as evident in Figs. 4F–4J. So, the GCN method is the preferable choice for medium-quality trajectories. For low-quality trajectories, neither the DBSCAN + Rules method nor the GCN method correctly classifies the input. For a trajectory displayed in Figs. 4K–4O that should be classified as "road", due to much noise generated by GNSS receivers, it is misclassified as "field" by both the DBSCAN + Rules and GCN methods, but it still can be correctly categorized by our TCModel-based DBSCAN+GCN method.

Meanwhile, this is also the first attempt to work on the balance problem in existing field-road classification methods. Our TCModel-based DBSCAN+GCN approach has been evaluated on a large-scale dataset collected at the peak of a harvesting season, a realistic agricultural scenario. As illustrated in Fig. 3, the time consumption of our TCModel-based DBSCAN+GCN is ranked second, demonstrating that the trajectory classification method can meet the balance requirement. Moreover, compared to the most effective method, GCN, our proposed approach achieves a remarkable reduction (65.20%) in time consumption with a small sacrifice of accuracy (0.17%) on the wheat harvester trajectory dataset (see Table 5). In a word, the TCModel-based DBSCAN+GCN has effectively addressed the issues of accuracy and efficiency presented in previous research and has achieved a good balance.

Overall, our TCModel-based DBSCAN+GCN can accomplish the field-road classification in a tolerable time consumption at the cost of an acceptable accuracy loss, particularly for large-scale trajectories. Therefore, it is a practical field-road classification method for real-time agricultural machinery management.

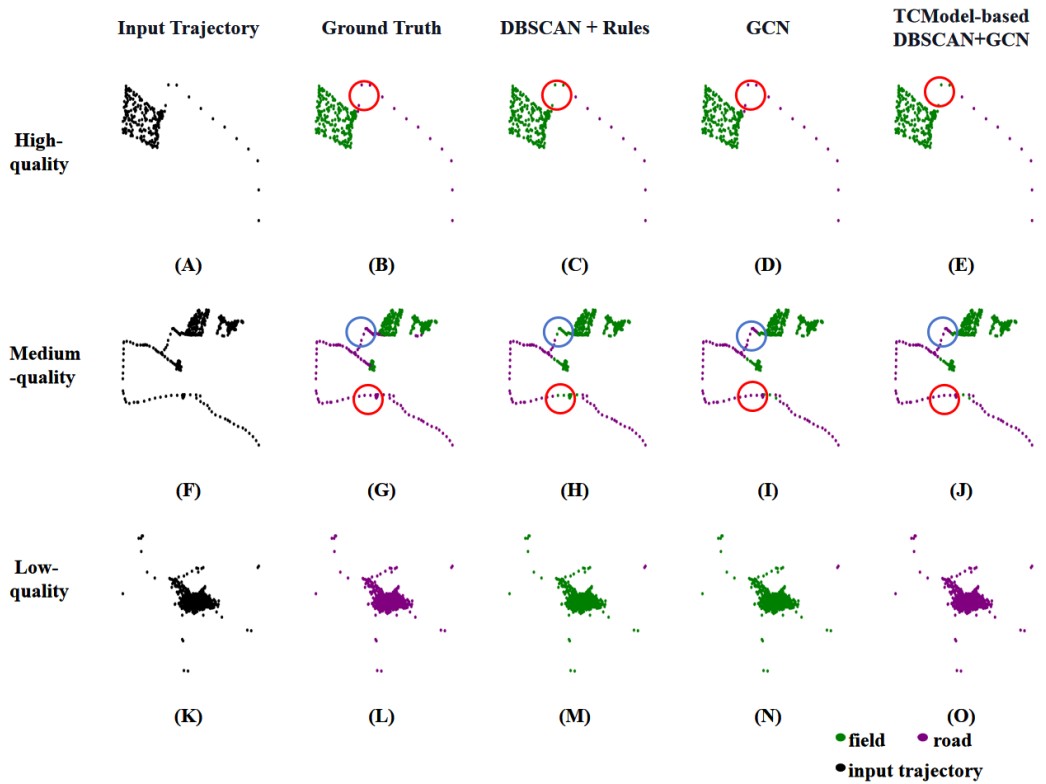

**Figure 4** **Comparison of field-road classification methods using various quality trajectories.** For (A–E), (A) is a high-quality trajectory input, and (B) is the ground truth trajectory, (C, D), and (E) represent the classified results by DBSCAN+Rules, GCN, TCModel-based DBSCAN+GCN, respectively. Points enclosed within circles in (B–E) indicate misclassified instances. The same analysis applies to (F–J) for a medium-quality trajectory and (K–O) for a low-quality trajectory.

## CONCLUSION

In this article, we propose a trajectory classification task to support efficient and effective field-road classification, which categorizes trajectories into three quality levels ("high-quality", "medium-quality", and "low-quality"). We then introduce an automated trajectory classification method that extracts 123 global features from the trajectory scope and 52 local features from the segment scope to predict the quality categories of input trajectories. Finally, we present the TCModel-based DBSCAN+GCN field-road classification method, which selects appropriate classification methods based on the qualities of input trajectories: DBSCAN + Rules for high-quality trajectories and GCN for medium-quality trajectories. The comprehensive experiments demonstrate that: (1) the automatic trajectory classification method has obtained promising results, outperforming the original method by 5.19% in F1-score and 1.89% in accuracy; (2) the TCModel-based DBSCAN+GCN has achieved sufficient efficiency with a manageable loss in accuracy, particularly when dealing with large-scale trajectories. Overall, the incorporation of the trajectory classification method can effectively balance the time consumption and accuracy of existing field-road classification methods.

The limitation of the proposed TCModel-based DBSCAN+GCN is that it is not an end-to-end architecture. There are two cascaded stages in the TCModel-based DBSCAN+GCN: classifying the quality of an input trajectory and classifying the agricultural machine's operational status within the trajectory. The two-stage design leads to the crucial information extracted during the field-road classification process can not be utilized for the trajectory classification process to improve its performance. To address this problem, in the future, we will explore an end-to-end architecture that can jointly learn both trajectory quality classification and field-road classification.

### Funding
The research was financially supported by the National Precision Agriculture Application Project (Grant/Contract number: JZNYYY001). The funders had no role in study design, data collection and analysis, decision to publish, or preparation of the manuscript.

### Grant Disclosures
The following grant information was disclosed by the authors:
National Precision Agriculture Application Project: JZNYYY001.

### Competing Interests
The authors declare there are no competing interests.

### Author Contributions
- Ying Chen conceived and designed the experiments, performed the experiments, analyzed the data, performed the computation work, prepared figures and/or tables, authored or reviewed drafts of the article, and approved the final draft.
- Kaiming Kuang conceived and designed the experiments, performed the experiments, analyzed the data, performed the computation work, prepared figures and/or tables, authored or reviewed drafts of the article, and approved the final draft.
- Caicong Wu conceived and designed the experiments, prepared figures and/or tables, and approved the final draft.

### Data Availability
The data is available at Zenodo: China Agricultural University. (2024). Field_road_dataset [Data set]. Zenodo. https://doi.org/10.5281/zenodo.10561232.

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
