# Peer review of "Trajectory classification to support effective and efficient field-road classification"

_PeerJ Computer Science, doi:10.7717/peerj-cs.1945_

## Round 0.1 · original submission · Minor Revisions

Based on the reviewers’ comments, you may resubmit the revised manuscript for further consideration. Please consider the reviewers’ comments carefully and submit a list of responses to the comments along with the revised manuscript.

**Language Note:** The review process has identified that the English language must be improved. PeerJ can provide language editing services - please contact us at [email protected] for pricing (be sure to provide your manuscript number and title). Alternatively, you should make your own arrangements to improve the language quality and provide details in your response letter. – PeerJ Staff

Reviewer 1 ·

Basic reporting

The paper under review presented a trajectory plan to improve the trajectory classification task. The authors presented three categories of trajectories namely high-quality, medium quality, or low-quality primarily based on point density, driving direction, and field boundary clarity in the trajectory. Though the approach presented in the paper improved the accuracy of the classification however, the authors need to compare the accuracy against each of the three categories i.e. high-quality, medium-quality, or low-quality.

Experimental design

The paper presented a detailed experimental design explaining the experimental setup and the dataset. Although the authors provided details about the methods however, the justifications why these methods have been selected are not clearly defined. I'd suggest the authors the present these missing details.

Validity of the findings

The authors presented a rigorous experimental evaluation of their approach. However, I'd suggest to evaluate their approach against all three categories separately. The discussion section is written well providing a convincing justification about the performance improvement of the proposed approach.

Reviewer 2 ·

Basic reporting

Positive remarks.
- The proposed study is fresh and significant.
- The paper is well structured and written.
- The methodology is original and it is described clearly
- The comparison done is important as it studied newly proposed work
- The conclusion support the obtained results

Suggested modifications:
** Check English mistakes
** The Abstract is too long please rewrite it and just summarize important information (don't give detailed ones)
** All abbreviations should only be used after their first definition (write each abbreviation in it full time when it is used for the first time; repeat that in the abstract and in the main text)
** Add a short paragraph at the end of the introduction that discribe how the rest of the paper is organized
** Add to your study the papers published in 2023 (and in 2024 as we are now at the end of 2023)
** enhance the qulaiti of figure 1
** can you explain what is relationship beteew the present work and your future one

Experimental design

As above

Validity of the findings

As above

·

Basic reporting

clear

Experimental design

Figures are not clear.

Validity of the findings

The word conclusions is wrong, just conclusion is sufficient

---

## Round 0.2 · accepted · Accept

Congratulations, the reviewers are satisfied with the revisions and the paper has been recommended for publication.

Reviewer 1 ·

Basic reporting

The authors have addressed all my suggested changes. The paper can be accepted for publication now.

Experimental design

No further comments.

Validity of the findings

No further comments.

Additional comments

No further comments.